# A Dataset of Annotated Omnidirectional Videos for Distancing Applications

**DOI:** 10.3390/jimaging7080158

**Published:** 2021-08-21

**Authors:** Giuseppe Mazzola, Liliana Lo Presti, Edoardo Ardizzone, Marco La Cascia

**Affiliations:** Dipartimento di Ingegneria, Università degli Studi di Palermo, 90128 Palermo, Italy; giuseppe.mazzola@unipa.it (G.M.); liliana.lopresti@unipa.it (L.L.P.); edoardo.ardizzone@unipa.it (E.A.)

**Keywords:** omnidirectional cameras, 360°, video dataset, depth estimation, distancing, video surveillance, tracking, pedestrian, equirectangular projection, spherical images

## Abstract

Omnidirectional (or 360°) cameras are acquisition devices that, in the next few years, could have a big impact on video surveillance applications, research, and industry, as they can record a spherical view of a whole environment from every perspective. This paper presents two new contributions to the research community: the CVIP360 dataset, an annotated dataset of 360° videos for distancing applications, and a new method to estimate the distances of objects in a scene from a single 360° image. The CVIP360 dataset includes 16 videos acquired outdoors and indoors, annotated by adding information about the pedestrians in the scene (bounding boxes) and the distances to the camera of some points in the 3D world by using markers at fixed and known intervals. The proposed distance estimation algorithm is based on geometry facts regarding the acquisition process of the omnidirectional device, and is uncalibrated in practice: the only required parameter is the camera height. The proposed algorithm was tested on the CVIP360 dataset, and empirical results demonstrate that the estimation error is negligible for distancing applications.

## 1. Introduction

Omnidirectional (or 360°) cameras have gained popularity in recent years, both for commercial applications and scientific goals. These devices can record a spherical view of an environment from every perspective, unlike traditional cameras that have a pre-set field of view. Commercial devices are now relatively cheap (a few hundreds of euros) and easy to use, as the acquisition process is fully automatic and there is no need to find the “right” perspective to take a picture from.

In commercial applications, users can typically interact with the recorded video, change the point of view, control the perspective, and feel surrounded by the environment. Moreover, 360° pictures and videos are available and usable on the most famous social networks (Facebook and YouTube) and can also be watched, besides using the regular 2D devices (laptops, mobile phones, and tablets), by using specific devices such as the head-mounted viewers (Google Cardboard, Oculus Rift, etc.) that help to improve the sense of immersivity of the users. From the scientific point of view, because of their features, omnidirectional devices find applications in virtual and augmented reality [1,2], mobile robotics [3,4], and video surveillance [5,6,7], which is also the topic of our work.

In this paper, we will present two new contributions to the research community: an annotated dataset of 360° videos that can be utilized for distancing problems, and a new approach to estimate the distances of objects to the camera. In the second section, we will describe the features of omnidirectional devices, showing their strengths and why they can be so convenient for computer vision applications. The third section will discuss the state-of-the-art datasets and techniques for depth estimation problems. In the fourth section, we will present our CVIP360 dataset and our distance estimation algorithm. The fifth section will describe the experimental results, and a conclusion section will end the paper.

## 2. Omnidirectional Cameras

Omnidirectional (or 360°) camera devices can acquire videos and panoramic images with a view (typically) of 360° horizontally and 180° vertically, resulting in a complete representation of the environment, and are particularly suitable for video surveillance, mobile robotics, and cultural heritage applications.

The first 360° devices were made from a traditional video camera filming a scene reflected on a mirror (usually of paraboloid shape) [8]. By knowing the mirror shape, it was possible to correct the distortion and reconstruct the original scene. These systems (reflectors) are not exactly omnidirectional (at least not vertically) because it is impossible to film everything that is above the mirror, and for this reason cameras must be placed high up inside the location. In the last years [9], the newest omnidirectional devices combine multiple calibrated cameras with partially overlapping fields of view. Each camera can shoot part of the scene, and the final image is reconstructed by stitching algorithms. These systems (diopters) are expensive because they need more calibrated cameras but can return high-resolution images and videos, albeit through computationally expensive reconstruction algorithms. On the other hand, the most recent devices typically use a system of two wide-angle lenses, each of which can shoot (more than) half of the scene. The entire panorama is reconstructed via software through stitching algorithms after correcting the distortion introduced by lenses if their shape is known. Since the system consists of only two cameras, it is relatively inexpensive, and the stitching process is much simpler and more efficient than when multiple cameras are used (Figure 1).

Regardless of the type of device, pixels of the images are mapped onto a sphere centered into the camera. To be displayed on the appropriate output devices (monitors, mobile phones, and head-mounted devices), equirectangular and cubic projections are often adopted [10]. Cubic projections allow for the displaying of images on a monitor or a viewer by mapping spherical points onto the plane tangent to the sphere. Often, a graphical interface allows for browsing of the cube. The equirectangular projection represents the whole sphere in a single image (2:1 ratio), in which the central row is the equator, and the upper and lower rows are the poles. Each row of the image corresponds to the intersection between the sphere and a plane parallel to the horizontal plane of the camera. This projection introduces a certain distortion, which is more visible approaching the poles. Similarly, each column of the image corresponds to the intersection between the sphere and a plane perpendicular to the horizontal one, passing through the poles and rotated by a given angle with respect to the polar axis.

## 3. Related Work

In this section, we will present the existing datasets of omnidirectional videos and the state-of-the-art techniques for depth estimation applications. In particular, the first subsection will focus on datasets for people-tracking applications, and the second one on those for depth estimation issues, together with the works in literature.

### 3.1. 360° Videos Datasets

Very few annotated datasets acquired by omnidirectional (360°) cameras are currently publicly available. Most of them are used to study the head movements of users watching 360° videos during immersive experiences; thus, annotations are generally provided for the users’ movements, attention, and eye gaze. A taxonomy of such datasets can be found in Elwardy et al. [11].

In some cases, 360° data is simulated from standard datasets. An example is the Omni-MNIST and FlyingCars datasets used in SphereNet [12].

In the field of visual tracking, recent works have adapted tracking techniques to 360° videos. Unfortunately, experimental results are often presented in private video collections or in YouTube videos for which annotations of the targets are not shared, for instance Delforouzi et al. [13]. In other cases, novel videos are instead acquired/downloaded from the internet and manually annotated. Mi et al. [14] propose a dataset including nine 360-degree videos from moving cameras. Three of them are captured on the ground and six of them are captured in the air by drones. Interesting targets such as buildings, bikers, or boats, one in each video, are manually marked in each frame as the ground truth of tracking.

In Liu et al. [15] two datasets, BASIC360 and App360, have been proposed. Basic360 is a simple dataset including seven videos acquired at 30 fps with a number of frames ranging between 90 and 408. In these videos, only two people are annotated. In App360 there are five videos of a length between 126 and 1234 frames and a number of annotated humans between four and forty-five. Data are acquired by a Samsung Gear 360.

In Yang et al. [16], a multi-person panoramic localization and tracking (MPLT) dataset is presented to enable model evaluation for 3D panoramic multi-person localization and tracking. It represents real-world scenarios and contains a crowd of people in each frame. Over 1.8 K frames and densely annotated 3D trajectories are included.

The dataset in Chen et al. [17] includes three videos, two acquired outdoors and one indoors. In the first two videos, there are five pedestrians. The dataset includes annotations for their full body (3905 bounding boxes) and, for some of them, their torsos. In the third video, only the full body of one pedestrian is annotated. Additionally, in this video, five objects, three heads, and six torsos are annotated. In the videos, pedestrians move randomly—alone or in groups—around the camera. All videos are characterized by strong illumination artifacts and a cluttered background. Annotations are provided in terms of bounding boxes detected on cubic projections. The dataset has been proposed to evaluate PTZ trackers where the PTZ camera is simulated based on the 360° videos.

Somewhat different is the dataset proposed by Demiröz et al. [18], where 360° cameras are mounted on the ceiling of a room and a sidewall; thus, there is no advantage of using a 360° camera to acquire the whole environment. Many datasets from fisheye cameras are available in the literature, especially for autonomous driving. For instance, WoodScape, presented by Yogamani et al. [19], is a dataset for 360° sensing around a vehicle using four fisheye cameras (each with a 190° horizontal FOV). WoodScape provides fisheye image data comprising over 10,000 images containing instance-level semantic annotation. However, to obtain a 360° view of the environment, several cameras are placed around the vehicle, which does not allow a unique sphere representing the environment and following the pinhole camera model to be obtained.

Table 1 summarizes the features of the discussed datasets, considering only those designed for visual tracking in 360° videos. Note that not all the features are available for all the datasets, as many of them are not (or are no longer) publicly available. With respect to those for which we have the complete information, our CVIP360 dataset is much larger, both in terms of the number of frames and number of annotated pedestrians; furthermore, the videos have a higher resolution.

Furthermore, differently from these datasets, our novel dataset focuses on distancing applications. It can also be used for tracking applications since pedestrians have been fully annotated but, differently than the above datasets, it also provides annotations of markers to assess distance computation of pedestrians to the camera.

Omnidirectional videos may be used as well in other computer vision research fields, such as face recognition. With respect to traditional techniques [20,21,22], the equirectangular projection introduces position-dependent distortions which must be considered when devising specific algorithms. This is a very new open issue, but some works have already been published [23,24]. To the best of our knowledge, there are no available datasets of videos acquired by an omnidirectional camera network (that is one of our future works), which are necessary in object re-identification applications [25].

### 3.2. Depth Estimation

Depth estimation is a well-known research field of computer vision [26] since the early 1980s, and has applications in various domains: robotics, autonomous driving, 3D modeling, scene understanding, augmented reality, etc.

Classical approaches are typically divided into two categories: active methods, which exploit sensors (e.g., the Microsoft Kinect) to build a map of the depths of the points in a scene, and passive methods, which use only visual information. The active methods have higher accuracy but cannot be used in all the environments and are more expensive. Passive methods are less accurate but have fewer constraints and are cheaper. Passive methods can be roughly divided into two subcategories, according to the hardware devices: stereo-based [27], in which a couple of calibrated or uncalibrated cameras are used to shoot a scene from two different points of view, and monocular [28], in which depth information is extracted by a single image. Some of the newest approaches use deep neural networks on monocular images [29,30,31,32,33,34].

Regarding 360° images and videos, some new works have been published in the very last years. Wang et al. [35] propose a trainable deep network, 360SD-Net, designed for 360° stereo images. The approaches of Jiang et al. [36] and Wang et al. [37] fuse information extracted by both the equirectangular and the cube map projections of a single omnidirectional image within a fully convolutional network framework. Jin et al. [38] propose a learning-based depth estimation framework based on the geometric structure of the scene. Im et al. [39] propose a practical method that extracts a dense depth map from the small motion of a spherical camera. Zioulis et al. [40] first proposed a supervised approach for depth estimation from single indoor images and then extended their method to stereo 360° cameras [41]. In Table 2 we show the features of the discussed datasets. With respect to the others, even if CVIP360 is a little bit smaller than two of them, CVIP360 is the only one that also includes videos of outdoor scenes and annotations of moving objects (pedestrians).

Almost all these methods are based on deep network techniques; they then need proper and solid datasets to train their frameworks. In the following sections, we will present a new annotated dataset of 360° videos that can be used for training supervised approaches, and an unsupervised algorithm to estimate the depths of the points in a scene.

#### 360° Datasets for Depth Estimation

Only a few existing datasets of omnidirectional videos for depth estimation applications are available to the research community. Matterport3D [42] is a dataset of 10,800 panoramic views of real indoor environments (with no pedestrians), provided with surface reconstructions, camera poses, and 2D and 3D semantic segmentation, designed for scene-understanding applications. Views are acquired by a rotating tripod system to build cylindrical projections. Stanford2D3D [43] includes 70,496 regular RGB and 1413 equirectangular RGB images, along with their corresponding depths, surface normals, semantic annotations, global XYZ OpenEXR formats, and camera metadata. Images of indoor environments, without persons, are acquired with the same setting of Mattertport3D and projected on a cylindrical surface. The 3D60 [40] dataset is created by rendering the two previous realistic datasets and two synthetic datasets, SceneNet [44] and SunCG [45], resulting in 23,524 unique viewpoints, representing a mix of synthetic and realistic 360° color, I, and depth, D, image data in a variety of indoor contexts. Even in this case, only environments and no pedestrians are acquired. PanoSUNCG [46] is a purely virtual dataset rendered from SunCG. This dataset consists of about 25k 360° images of 103 indoor scenes (with no pedestrians), panorama depth, and camera pose ground truth sequence. Table 1 summarizes the features of the discussed datasets.

## 4. CVIP360 Dataset

This section presents our CVIP360 dataset, which is freely available at the link [47]. The dataset is designed for testing tracking and distancing algorithms in omnidirectional videos. Our dataset includes 16 real videos, captured by a Garmin VIRB 360 omnidirectional device [48] in 4K (resolution 3840 × 2160) and with a frame rate of 25 fps. Ten of these videos have been shot indoors and six outdoors, to capture different luminance and topological conditions (Figure 2). The videos last from 15 to 90 s and show two to four people. Overall, the dataset includes 17,000 equirectangular frames and over 50,000 annotated bounding boxes of pedestrians (the product of the number of frames in a video and the number of persons). Each video has been annotated according to two different strategies:-We manually labeled all the pedestrians’ locations in the equirectangular frames by using bounding boxes.-We assigned to each pixel of the equirectangular frames a “real” distance to the camera’s location, according to a methodology described below in this section. In practice, we annotated only the image portion representing the area under the horizon. This area is related to the ground where people walk, as further explained in this section.

Note that in equirectangular images the horizon becomes a straight line across the middle of the image, and it is the projection of the camera horizontal plane onto the sphere. To our knowledge, this is the first dataset of omnidirectional videos of indoor and outdoor scenes, containing both information about the depths and the annotations of moving pedestrians, designed for both distancing and video surveillance applications.

### 4.1. Distancing Annotation

Before capturing the scene with our device, we put some markers onto the ground at fixed and known distances to the camera. For the outdoor videos, we put 10 markers, spaced one meter apart starting from the camera, at 0° (the direction in front of one of the two lenses) and 10 more markers at 90° (orthogonally, where both of the two lenses acquire information). In our device, each of the two lenses has a field of view of 201.8°, to avoid possible “blind spots” in the acquisition process. Finally, in the resulting stitched image the information around 90° and 270° is reconstructed by blending the two images coming from the camera.

Due to the acquisition and projection processes, all points of a horizontal plane equally distant to the camera in the 3D world belong to the same circle on that plane and correspond to horizontal lines in the equirectangular image (Figure 3). In our experimental setting, markers on the ground lie on a horizontal plane. Theoretically, we just needed to place markers along one direction, e.g., in front of the camera, since distances in the other directions can be inferred. We decided to also place markers along the orthogonal direction to measure distortions due to non-perfect camera alignment (in terms of pitch and roll). In such a case, the ground cannot be considered parallel to the camera horizontal plane, breaking our horizontality hypothesis. As a result, two markers at the same distance to the camera must match pixels in the same row of the equirectangular frame (unless for human errors in positioning them onto the ground, different ground levels, etc.). Small deviations from 0° in the roll or pitch camera angles may shift the marker coordinates in the images upwards or downwards by tens of pixels. These distortions have been corrected in post-processing by the VR distortion correction tool of the Adobe Premiere CC commercial software, to ensure the “horizontality” condition. The above procedure yields to the pixel coordinates of these markers, at fixed and known distances, in the equirectangular images.

On the other hand, for a more precise estimation of these distortions, in the case of indoor videos, we put markers along four directions (0°, 90°, 180°, and 270°), again spaced one meter apart, starting from the camera and reaching the furthest distances of the location, e.g., the walls, except in one direction, in which the site is partially open (see Figure 2). Due to poor illumination conditions and camera resolution, not all markers are clearly visible in the frames to match them to the pixels in the equirectangular image. For indoor videos, we decided to limit the maximum annotated distance to the camera to 6 m, which is adequate for most indoor environments. As well as the outdoor ones, these videos needed a post-processing step to correct distortions due to misalignments.

Knowing the relationships between pixel coordinates of markers in the images and their distance to the camera, all other matches between image pixels and true distances may be estimated by interpolation. To estimate this relationship, we tested several regression models, but we observed empirically that piecewise linear interpolation is accurate enough for our goals. Finally, we assigned a “real” distance to the camera to the pixels in the equirectangular image either by direct measurement (in the marker spots) or by interpolation (for all the other points) (Figure 4). Note that, with our method, we annotate only the lower part of the image (under the horizon), which includes the points on the ground satisfying the horizontality constraint. Note also that the angular coordinate of a point in the scene (pan) is the angle formed by the line connecting the point to the center of the camera, and the reference direction (e.g., the frontal one) is always known in omnidirectional cameras as:(1)$$\beta =\frac{x-\frac{w}{2}}{\frac{w}{2}}\cdot 180{}^{\circ}$$
where *w* is the image width and *x* is the *x*-coordinate of the equirectangular image. Knowing the distance and the pan angle of a point, we know its relative position to the camera in the real world (Figure 5).

## 5. Distance Estimation Algorithm

In this section, we describe our algorithm to estimate the distances of the objects in the scene to the camera. Our method is based on pure geometrical facts under the “horizontality” hypothesis discussed in the previous section. Furthermore, our method is uncalibrated, as there is no need to know or estimate the camera parameters, except for its height, which can be easily measured during the acquisition process.

Given an object in the scene (e.g., a person) touching the ground (supposed to be parallel to the camera horizontal plane) in a point (Figure 6), the distance, d, of this point to the camera can be estimated as:(2)$$d=h_{C}cot\alpha$$
where *h_C_* is the camera height and *α* is the angle between the line that joins the center of the camera and the horizon and the segment that joins the center of the camera and the point of intersection between the subject and the ground. Distance, *d*, is one of the two catheti of a right triangle and is related to the other one by the trigonometric Formula (2).

*α* can be computed directly from the frames of the videos as
(3)$$\alpha =\frac{y_{L}-\frac{h}{2}}{\frac{h}{2}}\cdot 90{}^{\circ}$$
where *y_L_* is the *y* coordinate, in pixels, of the lower bound of the subject (e.g., in the case of a person, the points where the feet touch the ground, that is the lower part of the bounding box) and h is the equirectangular image height (also in pixels) (Figure 7). Note that Equation (2) can also be applied if the height of the camera is greater than that of the subject. Remember that all the points in a plane parallel to the horizontal plane and equally distant to the camera are projected on the same row of the resulting equirectangular image. Finally, the only parameter that we need to know to compute the distances in the real world is the camera height, which can be easily measured during the acquisition phase.

Our method is totally unsupervised, does not need any training phase or any complex computation, but is just the result of a trigonometric formula. It can be used as a post-processing method of computer vision applications, e.g., tracking people, without affecting the computational load. For example, picking from the equirectangular image the angular coordinate of an object, as explained in Section 4.1, and estimating its distance from the camera, we know the location (in polar coordinates) of the object in our coordinate system (note that the camera location is the origin of the system). With simple mathematical operations, it is possible to compute the mutual distance between two objects in the real world, which is useful for many different applications, such as monitoring health restrictions, observing tourist behavior in cultural sites, etc.

## 6. Results

We tested our distance estimation algorithm on the CVIP360 dataset described in Section 3. The method could not be tested on other publicly available datasets since the camera height was unknown. Furthermore, in many of these datasets, videos are obtained by cylindrical projections, so our method, which is based on geometrical facts upon the spherical acquisition process of the scene, cannot be applied.

For each person and each frame in a video, we consider the vertical coordinate of the lower bound of the corresponding bounding box, which are points located on the ground. We can assign to the person a distance according to the graph in Figure 4, which relates bounding box coordinates to the real distance to the camera. This value is used as the ground truth value to assess the estimation error committed by our algorithm according to Formula (2). To evaluate the performances, we used two different metrics:

Mean Absolute Error
(4)$$MAE=\frac{1}{N}\underset{i}{\sum }\left|d_{e}^{i}-d_{T}^{i}\right|$$

Mean Relative Error
(5)$$MRE=\frac{1}{N}\underset{i}{\sum }\left|\frac{d_{e}^{i}-d_{T}^{i}}{d_{T}^{i}}\right|$$
where $d_{e}^{i}$ is the estimated distance, $d_{T}^{i}$ is the true distance, and *N* is the number of evaluated bounding boxes, which depends on the number of peeople and the number of frames in each video.

We measured these metrics only within the distance range for which we have the ground truth (10 m for outdoor videos, 6 m for indoor ones). All values above these thresholds are clipped to the maximum value. If both the estimated and the true distances are above the maximum distance, the corresponding error is not computed in the metrics.

MAE measures the absolute error (in meters) between the estimated distance of a person to the camera and the real one. Figure 8 shows how the MAE changes for increasing values of the ground truth distance in outdoor and indoor videos. Note that the average absolute error is 5 cm in outdoor videos and 2.5 cm in the indoor ones and that the error slightly increases only at higher distances, but is however very low and acceptable for any type of distancing application. Also note that moving away from the camera, points of the equirectangular image are closer to the horizon, and the space between a marker and another (1 m in the real world) is projected in a smaller range of pixels in the image.

MRE measures the relative error between the estimated distance of a person to the camera and its ground truth value, with respect to the last one. In Figure 9 we show the values of MRE versus the real distance values. As expected, MRE has its maximum value for the closest distance (as the estimated distance is normalized to the real distance), but it is in any case lower than 9% and, in most cases, around 1%. Similar to MAE, MRE also slightly increases at the furthest distances.

We also measured the MAE and MRE for different values of camera height, which is the only setting parameter that can affect our results (Figure 10). More specifically, assuming h_t_ is the true camera height, we measured how the error varies when a different value of camera height, h_e_, is used in Equation (2), thus simulating some possible inaccuracy in measuring this parameter during the video acquisition process.

MRE shows if the error is dependent on the value of the distance. By exploiting Equation (2), it can easily be shown that MRE depends only on the deviation between the true camera height and the value used to compute the estimated distance in Equation (2):(6)$$MRE=\frac{1}{N}\underset{i}{\sum }\left|\frac{d_{e}^{i}-d_{T}^{i}}{d_{T}^{i}}\right|=\frac{1}{N}\underset{i}{\sum }\left|\frac{h_{e}^{i}cot\alpha_{T}^{i}-h_{T}^{i}cot\alpha_{T}^{i}}{h_{T}^{i}cot\alpha_{T}^{i}}\right|=\frac{1}{N}\underset{i}{\sum }\left|\frac{h_{e}^{i}-h_{T}^{i}}{h_{T}^{i}}\right|$$
where $d_{e}^{i}$ is the estimated distance for the i-th point, $d_{T}^{i}$ is the corresponding ground truth value, and *α_T_* is the angle computed on the equirectangular image as explained in Equation (2). Therefore, a variation of 5% in the camera height parameter with respect to its true value causes a change of 0.05 in the MRE. The experimental results (see Figure 10) confirm this observation.

### Limitations and Strengths

In this section we present weaknesses and points of strength of our estimation method. In regard to the limitations, we believe that the accuracy of the method mainly depends on human imprecision in the setting phase (as the markers may all be placed incorrectly at the desired distances) and on the environment conditions (i.e., the ground is not exactly flat), rather than on the used equation.

We also observed that the error increases when the person appears close to the image borders (see Figure 11). This may depend on several reasons:-The lens distortion is at its maximum at the borders, so the assumption that the projection is spherical may not hold;-Pixels near the borders of the equirectangular image are reconstructed by interpolation;-After correcting the distortion to ensure the horizontality hypothesis, as explained in Section 4.1, a residual misalignment between the markers at the center and those at the borders results in different pixel positions.

With respect to the last point, in fact, we used as ground truth the average value of the pixel coordinates of the markers at the same distance, but the difference between the real distance and the estimated one, in practice, should depend on the object position. For simplicity, since this difference is very small in all the cases, we decided to ignore this effect.

Figure 12 also shows two representative cases where our algorithm cannot be applied. If a person jumps high, during the action for some instants the feet do not touch the ground, then the related bounding box is above the plane and the horizontality hypothesis does not hold. The hypothesis is broken as well when a person goes upstairs, as the feet are not in contact with the ground. In our dataset the number of frames in which this condition does not hold is negligible.

The strengths of the proposed method are its simplicity (it requires the use of just a single mathematical equation) as well as its efficiency, since no complex computation is required. The method is unsupervised (no training images are needed) and practically does not require calibration (only the camera height value has to be known). It can be used in several practical situations, as the method requirements are satisfied in most of the environments and scenes. Furthermore, results show the accuracy is very high, in most cases only a few centimeters of error.

## 7. Conclusions

Computer vision research opened up to the 360° multimedia processing world, and with this paper we want to give our contribution to the field. First, we contribute our dataset of 360° annotated videos, as currently only a few are available to the community. No technological improvements can be done without data to test new solutions, and our dataset will help researchers to develop and evaluate their works, especially for video surveillance applications, e.g., tracking pedestrians in 360° videos. To our knowledge this is the first dataset with indoor and outdoor omnidirectional videos of real scenarios, containing both depth information and annotations of moving pedestrians, designed for depth estimation and video surveillance applications. Second, we contribute our distance estimation method, which can be applied to a lot of case studies: for monitoring the physical distance between people in specific environments, for health regulations; to study the behavior of visitors in a place of interest, for tourist goals; in robotics, to help moving robots to plan their path and avoid obstacles; and in augmented reality applications, to improve user experiences. Despite some occasional cases in which the method cannot be applied, our approach is very simple, precise, and efficient, and does not need any complex calibration process to work.

In our future work, we plan to study and implement innovative techniques to track people in omnidirectional videos, starting from our dataset, analyzing the specific geometric properties of the acquisition process and the different projection methodologies. Furthermore, we plan to expand our dataset by acquiring scenes in different environments and conditions in addition to using a multicamera setting.

## Figures and Tables

**Figure 1 jimaging-07-00158-f001:**
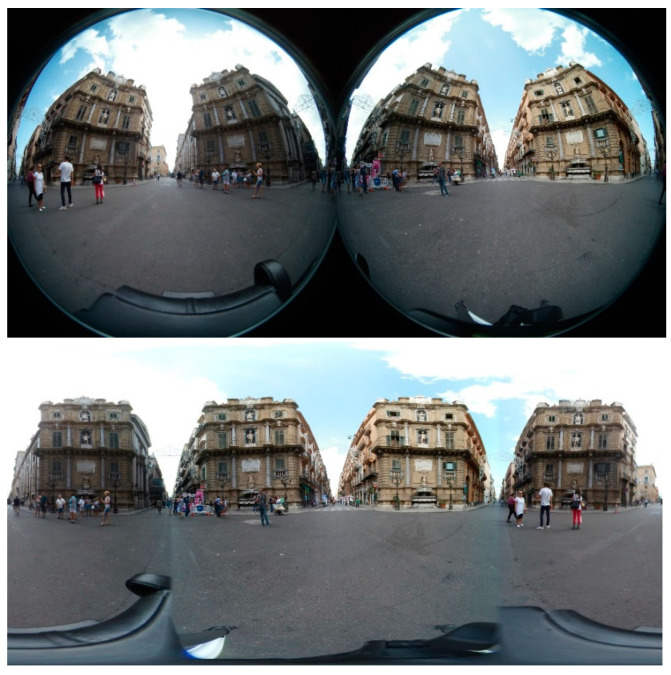
A scene acquired by a dual-lens 360° camera (Gear 360 by Samsung, Seoul, Korea) before (**top**) and after (**bottom**) stitching. The bottom figure is the equirectangular projection of the spherical acquisition. “Quattro Canti”, Palermo.

**Figure 2 jimaging-07-00158-f002:**
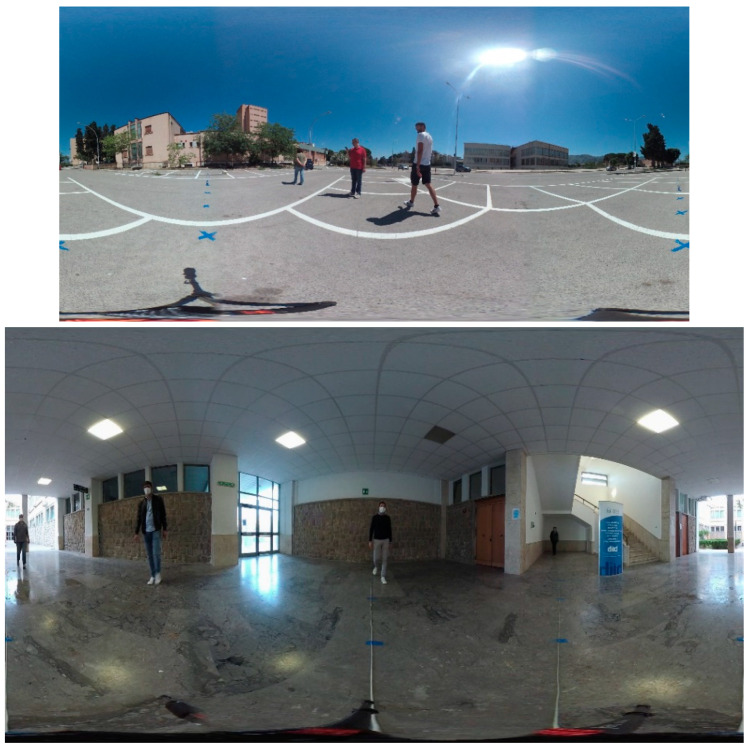
Two frames from the videos of the CVIP360 dataset. Outdoors (**top**) and indoors (**bottom**).

**Figure 3 jimaging-07-00158-f003:**
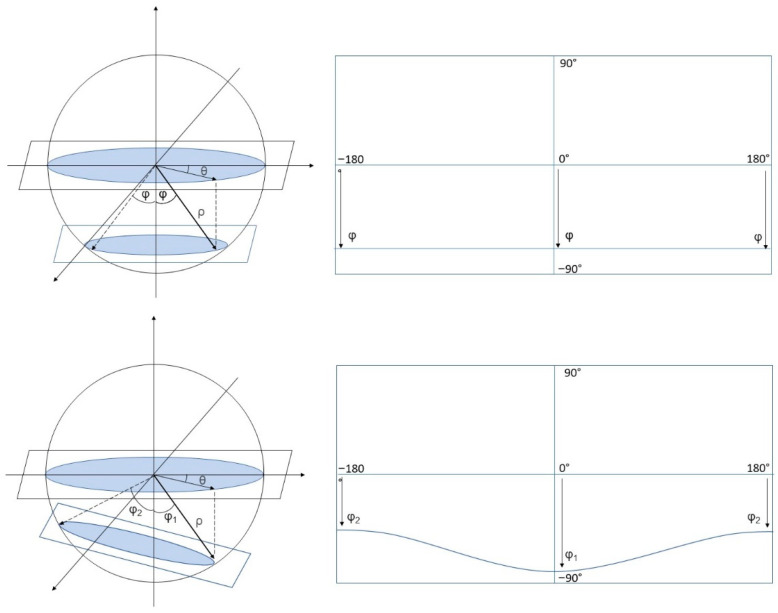
Projection of a circle onto an equirectangular image: in the case where the circle is on a plane parallel to the camera horizontal plane (**top**), and in the case where the two planes are not parallel (**bottom**). Note that in the second case points that belong to the same circle relate in the equirectangular image to a sinusoid-like curve.

**Figure 4 jimaging-07-00158-f004:**
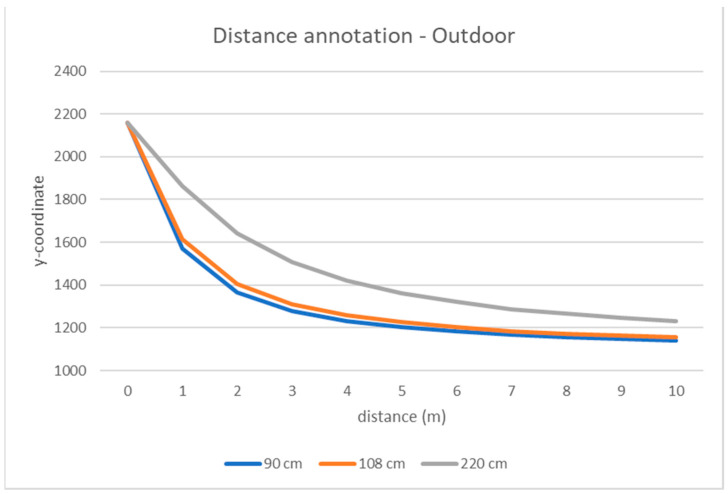
Relationship between the markers’ distance to the camera (in meters) and the corresponding y-coordinate in the equirectangular image, with different values of camera height (colored curves) for outdoor (**top**) and indoor (**bottom**) videos. Points at distances 1, 2, 3 … meters relate to the markers’ locations. Points between two markers are linearly interpolated. The points at the “zero” distance refer to the pixels belonging to the last row of the image (y = 2160).

**Figure 5 jimaging-07-00158-f005:**
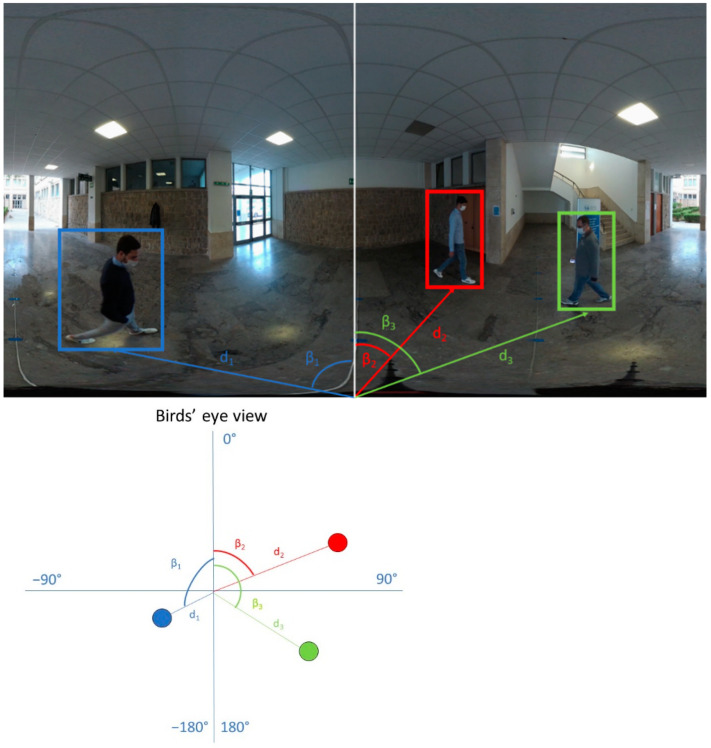
Annotation of pedestrians into the equirectangular image (**top**). Bird’s eye view of the same scene (**bottom**). For each pedestrian, the image shows the corresponding distance, *d*, on the ground and the angle, β (pan).

**Figure 6 jimaging-07-00158-f006:**
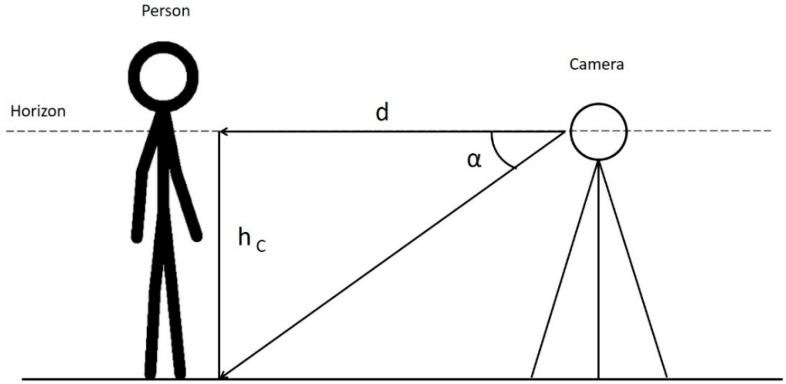
Geometrical representation of a subject acquired by a 360° device. *d* is the distance between the camera and the subject, *h_C_* is the camera height, and α is the angle between the line that joins the center of the camera and the horizon and the segment that joins the center of the camera and the point of intersection between the subject and the ground.

**Figure 7 jimaging-07-00158-f007:**
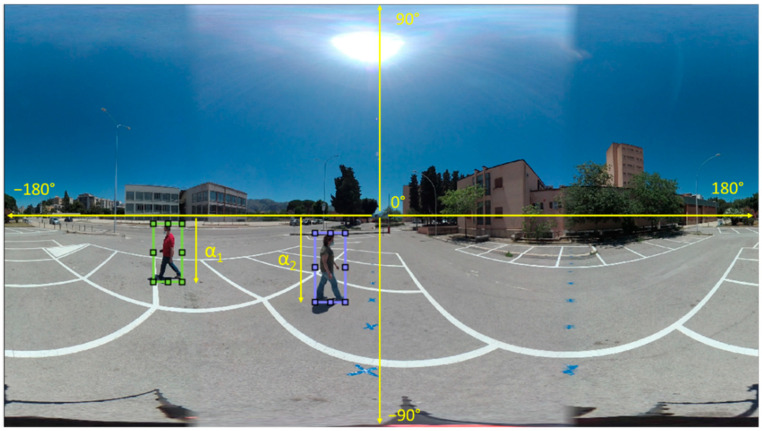
How to compute α from an equirectangular image. α is proportional to the segment that joins the horizon to the lower bound of the bounding box, as shown in Equation (3).

**Figure 8 jimaging-07-00158-f008:**
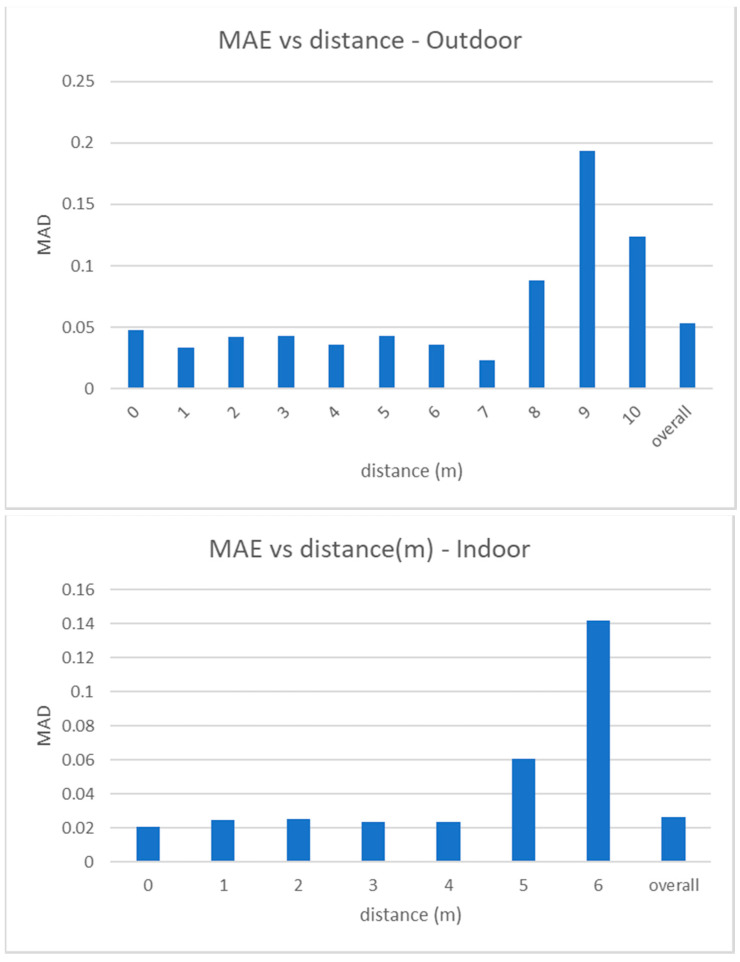
Experimental results. Mean absolute error versus distance to the camera, for outdoor (**top**) and indoor (**bottom**) videos. Note that the maximum measurable distance is 6 m in indoor videos and 10 m in outdoor videos. Graphs also report the overall values.

**Figure 9 jimaging-07-00158-f009:**
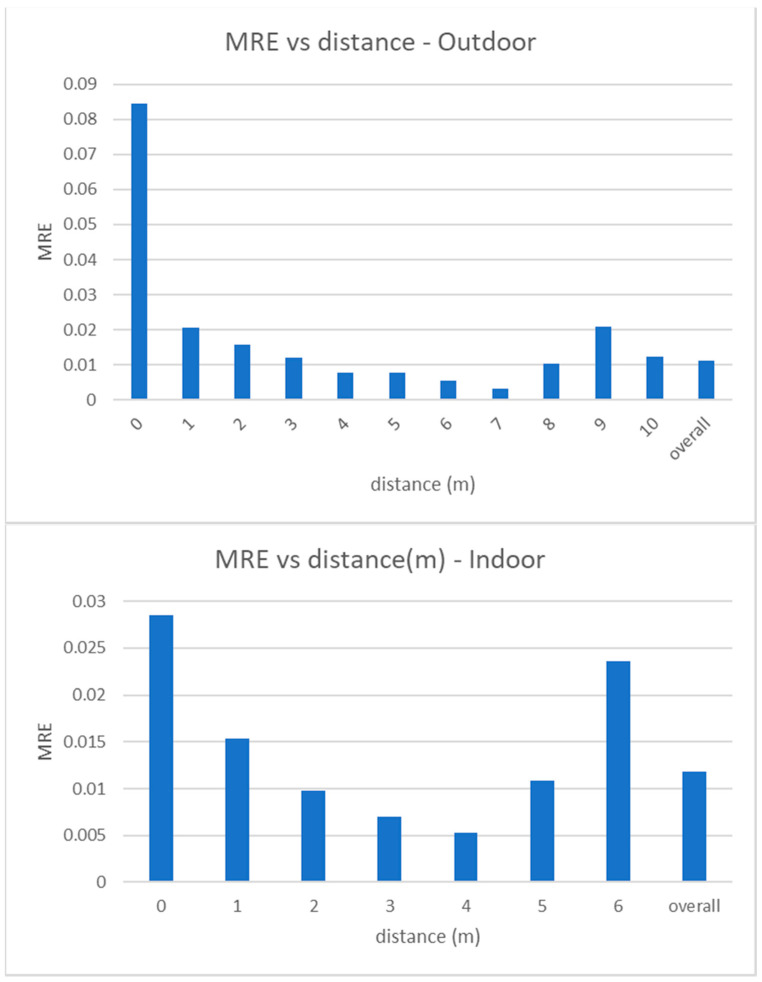
Experimental results. Mean relative error versus distance to the camera, for outdoor (**top**) and indoor (**bottom**) videos. Note that the maximum measurable distance is 6 m in indoor videos and 10 m in outdoor videos. Graphs also report the overall values.

**Figure 10 jimaging-07-00158-f010:**
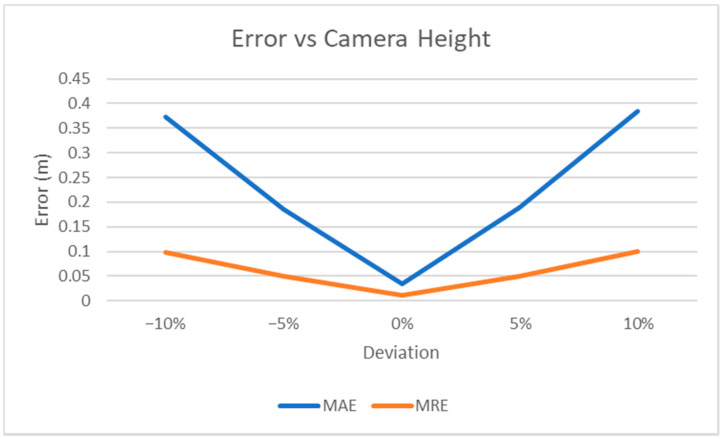
Experimental results. Sensitivity of MAE and MRE errors versus the height of the camera. x-axis is the deviation (in percentage) between the camera height set by the user in Equation (2) and the true value.

**Figure 11 jimaging-07-00158-f011:**
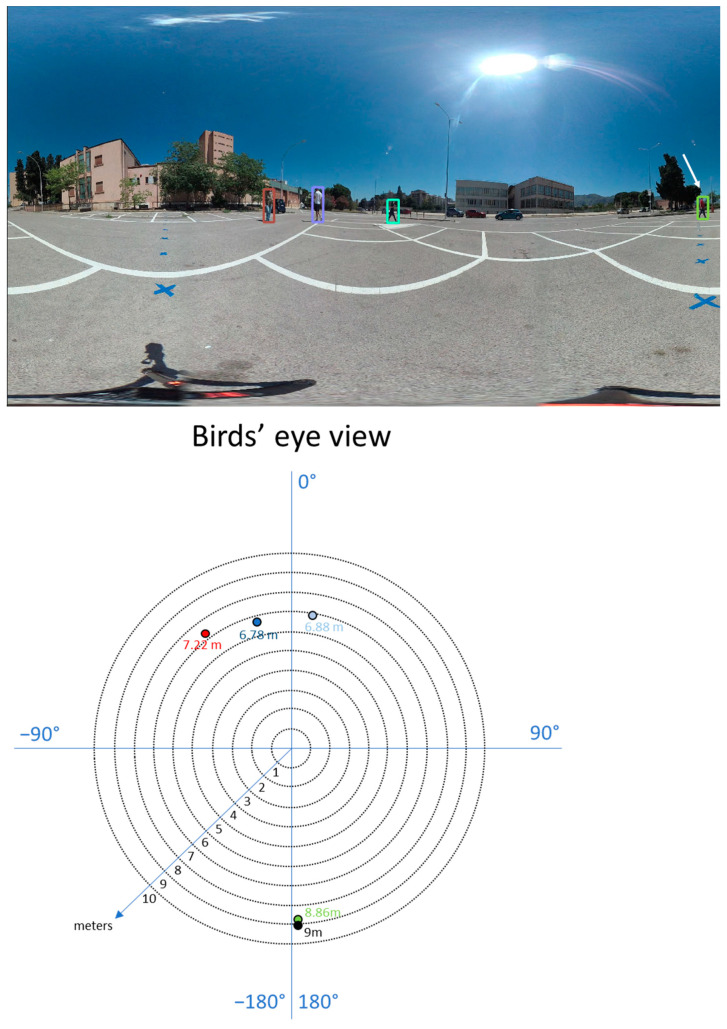
We observed that the method is less accurate when pedestrians approach the borders of the equirectangular image, as in the case in the figure (**top**). The bird’s eye view of the scene (**bottom**), showing the positions of all the persons into the camera system. For the person close to the borders, we show both the correct (according to the ground truth) and the estimated (according to our method) positions, in green and black respectively. In the other cases, the error is negligible with respect to the scale of the graph.

**Figure 12 jimaging-07-00158-f012:**
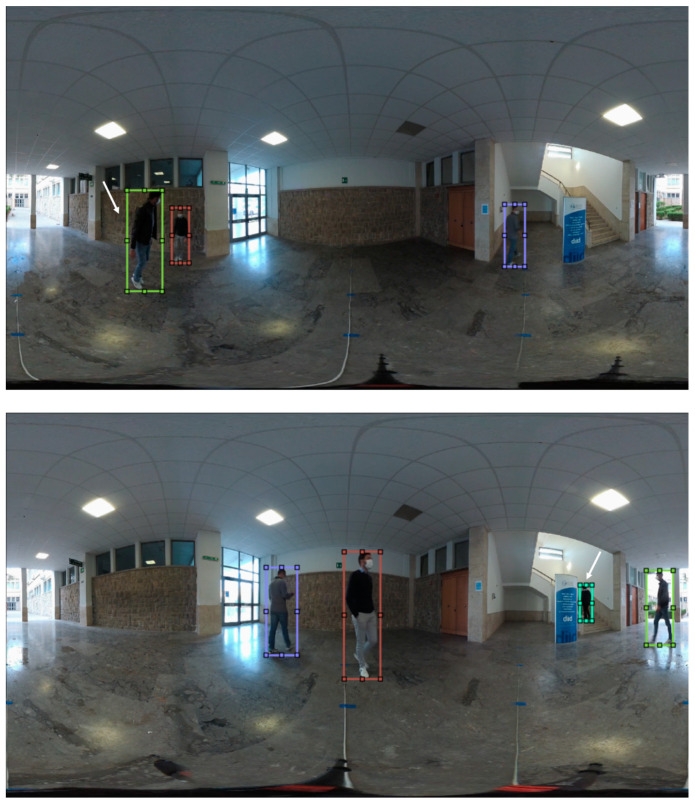
Some representative cases where the method cannot be applied: person jumping high (**top**); person going upstairs (**bottom**).

**Table 1 jimaging-07-00158-t001:** Summary of the features of the 360° datasets at the state of the art for visual tracking.

Name	Year	N° Videos	N° Frames	N° Annotations	Resolution	Availability
Delforouzi et al.	2019	14	n/a	n/a	720 × 250	Private
Mi et al.	2019	9	3601	3601	1920 × 1080	Public
Liu et al.	2018	12	4303	n/a	1280 × 720	Private
Yang et al.	2020	n/a	1800	n/a	n/a	Private
Chen et al.	2015	3	3182	16,782	640 × 480	Public
CVIP360 (ours)	2021	16	17,000	50,000	3840 × 2160	Public

**Table 2 jimaging-07-00158-t002:** Summary of the features of the state-of-the-art 360° datasets for depth estimation.

Name	N° 360° Images	Synthetic/Real	Environment	Pedestrians
Matterport3D	10,800	Real	Indoors	No
Stanford2D3D	1413	Real	Indoors	No
3D60	23,524	Real/Synthetic	Indoors	No
PanoSUNCG	25,000	Synthetic	Indoors	No
CVIP360 (ours)	17,000	Real	Indoors/Outdoors	Yes

## Data Availability

The CVIP360 dataset is publicly available for research purposes only and can be downloaded from: https://drive.google.com/drive/folders/1LMPW3HJHtqMGcIFXFo74x3t0Kh2REuoS?usp=sharing.

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
