# Peer review of "A Dataset of Annotated Omnidirectional Videos for Distancing Applications"

_2313-433X, 2021, doi:10.3390/jimaging7080158_

Round 1
Reviewer 1 Report
In this paper, the authors presented two main contributions: (1) a new annotated database of 360° videos, titled CVIP360, for distancing applications, and (2) a new method to estimate the distances of the objects in a scene from a single 360° image. In general, the paper is well written and organized, and the presented dataset can present an important impact to advance the research in video surveillance applications and industry. However, I have some recommendations in order to improve the quality of the manuscript:
- I recommend authors to add a comparative Table at the end of Section 3.1 that recapitulates the presented datasets described in this section, and what is the novelty of the presented dataset in comparison to the existing datasets.
- Couples of grammatical errors and misleading sentences have been found. It would be good if authors perform proofread by a native language expert.
- 16 videos are not sufficient to publish a new dataset, but we can accept the paper as preliminary work. So, it is recommended to augment the number of videos in future work.
- I recommend authors to add a brief discussion about the utility of the presented database/method of object estimation in the field of face recognition, and how they can solve some open face recognition challenges, which are discussed in detail in the following two papers:
- Adjabi, A. Ouahabi, Benzaoui, and S. Jaques "Multi‐Block Color‐Binarized Statistical Images for Single‐Sample Face Recognition", MDPI Sensors, vol.21 (3), pp.728, January 2021, Doi: 10.3390/s21030728.
- Adjabi, A. Ouahabi, Benzaoui, and A. Taleb-Ahmed, "Past, Present, and Future of Face Recognition: A Review", MDPI Electronics, vol.9 (8), pp.1188, July 2020, Doi:10.3390/electronics9081188.
Reviewer 2 Report
The scientific goal of this paper is interesting and important. This article presents the CVIP360 dataset, an annotated dataset of 360° videos for distancing applications, and a new method to estimate the distances of the objects in a scene from a single 360° image. Authors explain that empirical results demonstrate that the estimation error is negligible for any kind of distancing applications. However, in my opinion, the results of the proposed method are partially sufficient. I would like to encourage authors to continue this work and extend the following issues:
- I would like to suggest to provide, in the results section of the manuscript, some results of the algorithm to validate completely the distance measurements. That is, the Authors must provide an explanation and show success and error cases (sequence of images) that have been obtained, for example for moving pedestrians.
- I would like to suggest also for the validation of the proposed model to present several trajectories of the pedestrians that are estimated by the distance measurement algorithm. So, the different experiments based on real trajectories will determine the real performance of the method. Moreover, I would like also to recommend, to demonstrate the results in scenarios with complex and intermediate cases (for example with challenge cases like: low illumination, shadows, occlusions, complex background, etc.).
- The authors should show some examples where the proposed model fails explaining why.
- Please, authors must explain also in the manuscript a frank account of the strengths and weaknesses of the proposed model.
- Please, authors must also revise the link between the number of each reference and the reference in the manuscript.
Then, one of the fundamental issues with this paper is the lack of a complete study to demonstrate the advantages of the presented model to estimate the distances of the objects in a scene from a single 360° image. I would like to encourage authors to continue this work and extend the result and conclusion sections, to validate that the presented approach is precise and efficient in a real application like the trajectory of moving pedestrians.
Round 2
Reviewer 2 Report
In the revised manuscript, the descriptions and comments that I pointed out at reviewing have been modified sufficiently.